# Secure Diffusion Model Unlocked: Efficient Inference via Score Distillation

## Abstract

As services based on diffusion models expand across various domains, preserving the privacy of client data becomes more critical. Fully homomorphic encryption and secure multi-party computation have been employed for privacy-preserving inference, but these methods are computationally expensive and primarily work for linear computations, making them challenging to apply to large diffusion models. While homomorphic encryption has been recently applied to diffusion models, it falls short of fully safeguarding privacy, as inputs used in the $\epsilon$ prediction are not encrypted. In this paper, we propose a novel framework for private inference for both inputs and outputs. To ensure robust approximations, we introduce several techniques for handling non-linear operations. Additionally, to reduce latency, we curtail the number of denoising steps while minimizing performance degradation of conditional generation through score distillation from the unconditional generation of the original model with full denoising steps. Experimental results show that our model produces high-quality images comparable to the original, and the proposed score distillation significantly enhances performance, compensating for fewer steps and approximation errors.

## 1 Introduction

Modern conditional diffusion models show impressive performance in generating images and are employed across a wide range of applications. Notably, Stable Diffusion (Rombach et al., 2022), one of the leading models, can produce high-quality images from user prompts or visual inputs such as sketches and key points. Building on the recent success of diffusion models, there is an increasing interest in their application in privacy-sensitive fields, such as medical domain and content creation. In the medical field, they are utilized for translating MRI to CT, denoising medical images, and medical anomaly detections. Additionally, DALL·E (Ramesh et al., 2022) and Midjourney have become popular as commercial tools for content generation and creative industries.

Despite these advancements, the substantial computational costs of image generation typically necessitate processing on powerful remote servers rather than local devices. This reliance on external servers introduces potential privacy concerns, as both user inputs and corresponding outputs may be exposed to the model provider. These concerns are especially critical in the medical domain, where inputs consist of sensitive information such as patient profiles and medical images, and outputs also include critical data such as CT scans in image translation tasks or enhanced images in denoising tasks. Even when inference is performed on client devices, the risk of exposing the model provider's proprietary model weights remains a critical issue, further complicating the deployment of diffusion models in privacy-sensitive fields.

Faced with the issue, several studies utilize secure computation methods such as fully homomorphic encryption (FHE) (Gentry, 2009) and secure multi-party computation (MPC) (Evans et al., 2018). CryptoNet (Gilad-Bachrach et al., 2016) propose a HE-friendly architecture for convolutional neural networks, using polynomial activations. MPCFORMER (Li et al., 2023) introduces a framework for private inference in transformer models by leveraging secure MPC. However, it is challenging to utilize these approaches for large diffusion models. First, both FHE and MPC require substantial overhead since FHE has significant computational costs and secure MPC introduces significant communication costs. Additionally, both FHE and MPC only support linear operations, necessitat-

ing the approximation of non-linear functions like GroupNorm (Wu & He, 2018), LayerNorm (Ba, 2016), SiLU, GeLU (Hendrycks & Gimpel, 2016), and softmax.

Recently, HE-Diffusion (Chen & Yan, 2024) introduces FHE to secure outputs during the denoising process. At each denoising step, the client transmits both the intermediate representations of the text input and the distorted image to the model provider. Then, the model provider predicts the noise, adds it to the encrypted part, and generates the next-step image. A key advantage of this method is that noise prediction occurs outside the ciphertext space, avoiding the need to approximate non-linear functions and significantly reducing latency caused by the computations in the ciphertext space. However, this approach still has a risk of privacy leakage: the text inputs remain unencrypted, and Chen et al. (2024) observes that it may be possible to recover the text inputs from the intermediate embeddings. Therefore, encrypting the text inputs is essential to ensure complete privacy.

To safeguard the privacy of both inputs and outputs, we propose Private Inference for Diffusion Models (PIDM) that enables image generation from encrypted inputs using approximation methods. Inference in the ciphertext space for models like Stable Diffusion poses challenges, due to the numerical instability associated with approximating non-linear operations, such as GroupNorm, SiLU, GeLU, and softmax. To address this, we introduce the techniques that provide more stable approximations for normalization, the error functions, and softmax. Through the experiments, we show that our approach performs better than the existing approximation used in CrypTen (Knott et al., 2021).

Furthermore, to reduce the latency, we propose a novel score distillation sampling approach to guide the conditional denoising process using the unconditional path. The key idea is to leverage computations in plaintext space to minimize those in ciphertext space, thereby reducing latency with only a small performance trade-off. Specifically, our method takes fewer denoising steps for the conditional denoising process and this strategy would result in performance degradation. However, by using the unconditional denoising process, which takes the full number of steps and does not utilize the approximations, as a guiding score, we mitigate this issue. Empirical results show that this approach significantly reduces latency with only a minor decrease in performance.

Our contribution is threefold:

- To the best of our knowledge, we are the first to propose a framework for Stable Diffusion that generates images from encrypted inputs, utilizing several techniques.
- We introduce a novel score distillation sampling method from the unconditional denoising process, reducing computations within the ciphertext space by taking more computational effort to the plaintext space.
- Empirical results demonstrate that our framework achieves comparable performance to the original Stable Diffusion, while our sampling method reduces the latency with small performance trade-offs.

## 2 PRELIMINARIES

### 2.1 DIFFUSION MODEL

Diffusion models (Sohl-Dickstein et al., 2015) learn data distribution $p_{\text{data}}$ by denoising a variable following a normal distribution. To learn the reverse process, the models are trained to produce the latent variables from noise over $T$ steps. Let $\mathbf{x}_1, \ldots, \mathbf{x}_T$ be latent variables where the initial noise, $\mathbf{x}_T$, follows Gaussian distribution and $\mathbf{x}_0 \sim p_{\text{data}}$. For the model parameter $\theta$, the previous variable is obtained by the reverse process as follows:

$$p_\theta(\mathbf{x}_{t-1}|\mathbf{x}_t) := \mathcal{N}\left(\mathbf{x}_{t-1}; \mu_\theta\left(\mathbf{x}_t, t\right), \boldsymbol{\Sigma}_\theta\left(\mathbf{x}_t, t\right)\right). \tag{1}$$

In DDPM (Ho et al., 2020), the mean and covariance, $\mu_\theta(\mathbf{x}_t, t)$ is computed as:

$$\mu_\theta\left(\mathbf{x}_t, t\right) = \frac{1}{\sqrt{\alpha_t}}\left(\mathbf{x_t} - \frac{\beta_t}{\sqrt{1 - \bar{\alpha}_t}}\epsilon_\theta(\mathbf{x}_t, t)\right), \boldsymbol{\Sigma}_\theta\left(\mathbf{x}_t, t\right) = \beta_t \mathbf{I}, \tag{2}$$

where $\beta_1, \ldots, \beta_T$ are variance schedules, $\alpha_t = 1 - \beta_t$, and $\bar{\alpha}_t = \prod_{s=1}^{t} \alpha_s$. For $\epsilon_\theta(\mathbf{x}_t, t) = \epsilon_\theta\left(\sqrt{\bar{\alpha}_t}\mathbf{x}_0 + \sqrt{1 - \bar{\alpha}_t}\epsilon, t\right)$, the diffusion model is trained by the $\epsilon$ prediction:

$$\mathbb{E}_{t,\mathbf{x}_0,\epsilon}\left[\|\epsilon - \epsilon_\theta(\mathbf{x}_t, t)\|_2^2\right]. \tag{3}$$

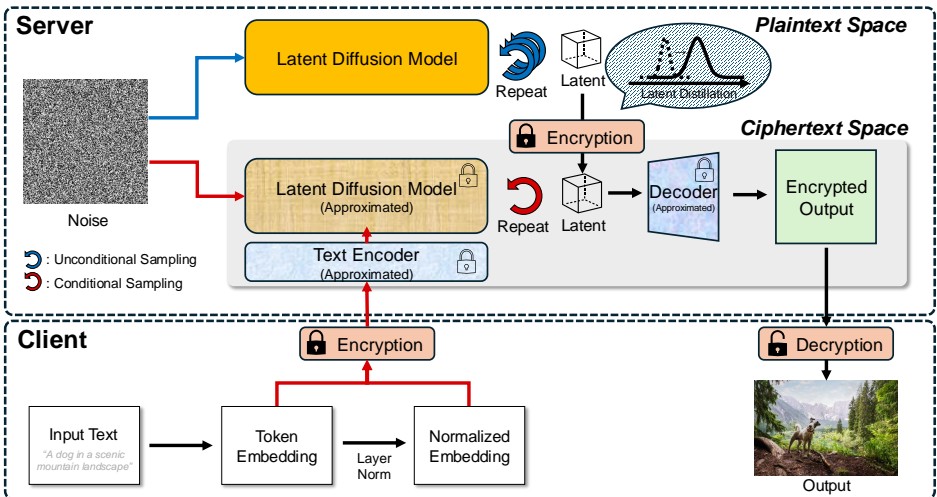

Figure 1: The overview of PIDM. The client sends encrypted input to the server, which processes and returns the result. Denoising process occurs in the ciphertext space, with the non-linear functions of the diffusion model approximated as linear, as only linear operations are possible in this space. Score distillation from unconditional generation is employed to reduce latency by decreasing the computations in the ciphertext space.

Note that we will utilize the score function as $s_\theta(\mathbf{x}_t, t) = -\epsilon_\theta(\mathbf{x}_t, t)/\sqrt{\beta_t}$ based on Tweedie's formula (Robbins, 1992). However, diffusion models are computationally intensive, as the diffusion process operates at the pixel level, resulting in substantial memory and computational demands, particularly for high-resolution image generation. To reduce the computational complexity, Latent Diffusion Model (LDM) (Rombach et al., 2022) performs diffusion processes in latent space using autoencoder. In the training process, diffusion models are trained by:

$$\mathbb{E}_{t,\mathbf{z}_0,\epsilon}\big[\|\epsilon - \epsilon_\theta(\mathbf{z}_t, t)\|_2^2\big], \tag{4}$$

where $\mathbf{z}_t$ is calculated by adding noises to the latent vector of the original image.

## 2.2 PRIVATE INFERENCE

Fully homomorphic encryption (FHE) (Gentry, 2009) and secure multi-party computation (MPC) (Evans et al., 2018) are widely adopted for private inference. FHE allows operations to be performed directly on encrypted data, ensuring that the decrypted result is nearly identical to the output of plaintext data. For input $x, y$, the following approximations hold:

$$\text{Dec}\left(\text{Enc}(x)\right) \approx x, \text{Dec}\left(\text{Enc}(x) \oplus \text{Enc}(y)\right) \approx x + y, \text{Dec}\left(\text{Enc}(x) \odot \text{Enc}(y)\right) \approx x * y, \tag{5}$$

where Enc and Dec denote the encryption and decryption functions, respectively. $(\oplus, \odot)$ are operations in the ciphertext space while $(+, *)$ correspond to operations in the plaintext space. The key issue here is that FHE supports only addition and multiplication. Furthermore, the computational overhead is significant (nearly $100\times$ slower than in the plaintext space) for large models due to bootstrapping required to mitigate decryption errors that arise from extensive computations.

On the other hand, secure MPC allows multiple parties to jointly compute operations on combined data while preserving the privacy of each party's data. In secure MPC, secret data is divided into multiple shares and distributed among participants (Damgård et al., 2012), ensuring that reconstructing the original data is computationally infeasible without all shares. Computations are executed on these shares, ensuring privacy is maintained throughout the process. However, secure MPC also supports only addition and multiplication and demands significant communication overhead.

In deep learning, these private inference methods are commonly used in machine learning-as-a-service scenarios. In such a scenario, a client submits encrypted data to a model provider, which processes it and returns the results. Both the data and model parameters are encrypted and operated on in the ciphertext space. Since non-linear operations are unsupported, methods such as polynomial approximations or iterative algorithms have been proposed (Gilad-Bachrach et al., 2016; Li et al., 2023). Other approaches focus on reducing computational costs by introducing new frameworks or protocols (Hao et al., 2022; Wu et al., 2024).

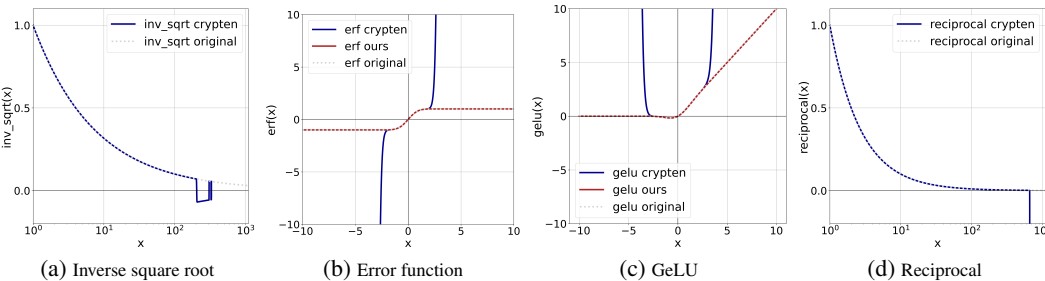

(a) Inverse square root  (b) Error function  (c) GeLU  (d) Reciprocal

Figure 2: The graphs highlight CrypTen's limitations compared to our approximation. Figures 2a and 2d show that CrypTen's approximations for the inverse square root and reciprocal functions become inaccurate for values above 100 and 400, respectively. Figure 2b contrasts CrypTen's use of Taylor approximation for the Gaussian error function with our approach, which employs the hyperbolic tangent function. Figure 2c compares the GeLU function, showing the original, CrypTen, and our method based on our error function.

## 3 PRIVATE INFERENCE FOR DIFFUSION MODEL

We now present our overall algorithm whose two key ingredients are i) approximation techniques for complex operations of diffusion models and ii) score distillation from unconditional generation. Before delving into our method, note that while we mainly use a Stable Diffusion (Rombach et al., 2022) for our explanations, our method is not inherently dependent on Stable Diffusion and can be applied to other models. In our framework, the client transmits encrypted input to the model provider, who processes the input and returns the result. To ensure private inference, either homomorphic encryption (FHE) or secure multi-party computation (MPC) can be employed. The overall framework is illustrated in Figure 1.

Specifically, the client computes the token embeddings and the first LayerNorm (Ba, 2016) within the text encoder of Stable Diffusion. These tensors are then encrypted and transmitted to the server (model provider). This step is performed client-side because the initial token embeddings have high variance, and processing them before encryption preserves performance. Since this computation represents a very small fraction of the overall workload, the client's computational burden remains minimal. After these initial operations, all subsequent layers and components are processed within the ciphertext space on the server.

To perform the denoising process in the ciphertext space, where only linear operations are supported, we approximate the non-linear functions in the diffusion model to linear counterparts. The details of these approximation techniques are discussed in Section 3.1. To reduce latency, we decrease the number of denoising steps. To address the score deviations caused by insufficient steps and the approximations, we apply score distillation from unconditional generation, as detailed in Section 3.2.

### 3.1 PRACTICAL APPROXIMATION TECHNIQUES

Stable Diffusion includes several non-linear functions such as GroupNorm (Wu & He, 2018), LayerNorm, SiLU, GeLU (Hendrycks & Gimpel, 2016), and softmax. However, fully homomorphic encryption (FHE) and secure multi-party computation (MPC) only support linear operations, necessitating the approximation of non-linear functions for private inference. The approximation methods utilized in CrypTen (Knott et al., 2021), a widely used open-source library, may not suffice to achieve comparable performance to the original model. To address this challenge, we suggest several techniques that enhance the approximation abilities of the library without additional training. Note that each technique is significant since removing any of them can lead to noisy or black images.

First, GroupNorm and LayerNorm normalize input tensors using the inverse square root of the variance, a non-linear operation that can be approximated with the Newton-Raphson method. However, we observe that the approximation error increases significantly for input values exceeding $10^2$ in Figure 2a. Using the fact that $\frac{\mathbf{x}-\text{mean}(\mathbf{x})}{\text{deviation}(\mathbf{x})} = \frac{\mathbf{x}/d-\text{mean}(\mathbf{x}/d)}{\text{deviation}(\mathbf{x}/d)}$ for any non-zero $d$, we scale down the variance of the input tensor by a sufficiently large factor before normalization. Specifically, we increase $d$ as the number of channels in the group decreases, leveraging the fact that variance is proportional to the number of samples. The details of scaling factors are provided in Appendix A.

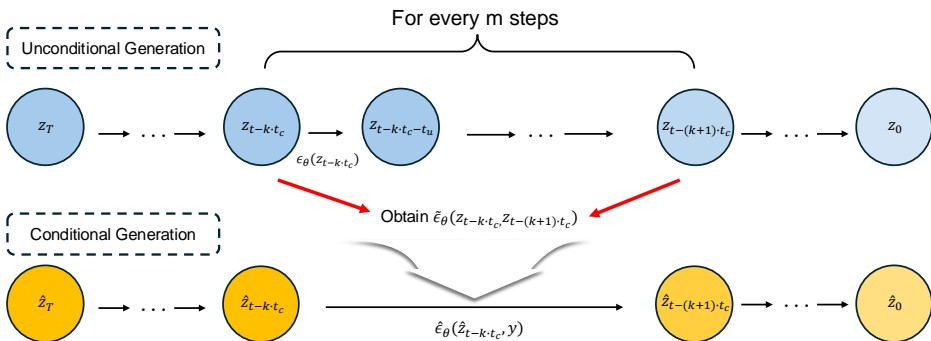

Figure 3: The overview of score distillation from unconditional generation. Conditional generation derives $\hat{\epsilon}_\theta(\hat{\mathbf{z}}_t, y, t)$ from the results of unconditional generation that have undergone full denoising steps, and utilizes it in its generation.

Next, SiLU contains the sigmoid, which can be approximated as reciprocal$(1 + \exp(-x))$. The exponential function is approximated as $\exp = (1 + \frac{x}{n})^n$ for a large value of $n$ in CrypTen. GeLU includes both the sigmoid and the Gaussian error function. The sigmoid is approximated similarly to SiLU, while the error function is approximated using a Taylor series in CrypTen. However, this high-order Taylor approximation can lead to explosive values for larger inputs in Figure 2b. To address this, we adopt an approximation based on the hyperbolic tangent function: $\tanh\left(\frac{2}{\sqrt{\pi}}(x + \frac{11}{123}x^3)\right)$. Since $\tanh(x) = 2 \cdot \text{sigmoid}(2x) - 1$ and the output of the sigmoid is bounded by $[-1, 1]$, this approximation is more robust for large input values, as shown in Figure 2b and 2c.

Lastly, softmax involves both exponential and reciprocal functions. We applied the same approximations as those used in SiLU. However, we observe that the reciprocal exhibits significant errors for input values exceeding 400 in Figure 2d. While CrypTen mitigates this issue by subtracting the maximum input value, as $\frac{\exp(x_i)}{\sum_j \exp(x_j)} = \frac{\exp(x_i - x_{\max})}{\sum_j \exp(x_j - x_{\max})}$ for $x_{\max} = \max_j x_j$, the sum of exponential values can still become large when processing many tokens. In Stable Diffusion, the model processes 4096 visual tokens, leading to substantial approximation errors. To alleviate this, we divide the input tensor by the square root of the number of tokens after applying the exponential but before applying the reciprocal. Thus, we compute the softmax as: $\frac{\exp(x_i - x_{\max})}{\sum_j \exp(x_j - x_{\max})} = \frac{\exp(x_i - x_{\max})/\sqrt{d}}{\sum_j (\exp(x_j - x_{\max}/\sqrt{d}))}$, where $d$ is the number of tokens. Additional minor techniques are discussed in Appendix A.

## 3.2 SCORE DISTILLATION FROM UNCONDITIONAL GENERATION

Since computations in the ciphertext space are drastically slower than in the plaintext space as mentioned in Section 2.2, it is critically important to perform only a limited number of denoising steps to reduce the inference overhead. However, the naive reduction in denoising steps, combined with approximation errors, can result in severe performance degradation. To address this, we propose score distillation from unconditional generation (SDU), which refines the score obtained from the conditional generation with unconditional generation of the original diffusion model with full denoising steps. The key insight is that, since unconditional generation can be performed in plaintext space, it can take more steps and avoid approximation errors, resulting in a distribution that is closer to that of the original model. The overall process is illustrated in Figure 3.

Specifically, our objective is to minimize the divergence between the image distribution generated by the encrypted model and that produced by the original model. Let $p_\theta(\hat{\mathbf{z}}_t|y)$ denote the image distribution of the original model and $\hat{p}_\theta(\hat{\mathbf{z}}_t|y)$ indicate that of the encrypted model. The widely adopted objective is computed as $D_{\text{KL}}(\hat{p}_\theta(\hat{\mathbf{z}}_t|y), p_\theta(\hat{\mathbf{z}}_t|y))$. Since our main goal is to enhance performance through a correction step during inference, we focus on reducing the divergence for high-density samples from the original model. Therefore, our objective is formulated as:

$$\min_{\hat{\epsilon}_\theta} D_{\text{KL}}(p_\theta(\hat{\mathbf{z}}_t|y), \hat{p}_\theta(\hat{\mathbf{z}}_t|y)). \tag{6}$$

Note that $\hat{p}_\theta(\hat{\mathbf{z}}_t|y)$ may deviate from $p_\theta(\hat{\mathbf{z}}_t|y)$ due to the limited number of steps and approximation errors, resulting in performance degradation. This phenomenon is illustrated in Figure 6 and Table 1.

---

**Algorithm 1** Score distillation from unconditional generation

---

1: **Input:** weight function $\mathbf{w}(t)$, learning rate $\eta$, condition $y$, number of total timestep $T$, timestep offset for unconditional generation $t_u$, for conditional generation $t_c = m \cdot t_u$, classifier-free guidance strength $w_{\text{cfg}}$

2: $\mathbf{z}_T \sim \mathcal{N}(\mathbf{0}, \mathbf{I})$

3: $\hat{\mathbf{z}}_T \leftarrow \mathbf{z}_T$

4: $\mathbf{n} \sim \mathcal{N}(\mathbf{0}, \mathbf{I})$ if $t > t_u$, else $\mathbf{n} = \mathbf{0}$

5: **for** $t = T - t_u, T - 2t_u, \ldots, t_u, 0$ **do**  ▷ Unconditional generation

6:     $\mathbf{z}_t \leftarrow \frac{1}{\sqrt{\alpha_{t+t_u}}}\left(\mathbf{z}_{t+t_u} - \frac{\beta_{t+t_u}}{\sqrt{1-\bar{\alpha}_{t+t_u}}}\epsilon_\theta(\mathbf{z}_{t+t_u}, t+t_u)\right) + \sqrt{\beta_{t+t_u}}\mathbf{n}$

7:     **if** $t \in [T - t_c, T - 2t_c \ldots, t_c, 0]$ **then**  ▷ Conditional generation

8:         $\epsilon_\theta^{\textbf{SDU}} \leftarrow \frac{\sqrt{1-\bar{\alpha}_{t+t_c}}}{\beta_{t+t_c}}\left(\mathbf{z}_{t+t_c} - \sqrt{\alpha_{t+t_c}}\mathbf{z}_t\right)$  ▷ Compute $\epsilon_\theta^{\textbf{SDU}}$ by reverse engineering

9:         $\hat{\epsilon}_\theta^{\text{cfg}} \leftarrow (1 + w_{\text{cfg}})\hat{\epsilon}_\theta(\hat{\mathbf{z}}_{t+t_c}, y, t+t_c) - w_{\text{cfg}}\hat{\epsilon}_\theta(\hat{\mathbf{z}}_{t+t_c}, t+t_c)$

10:        $\hat{\epsilon}_\theta^{\text{cfg}} \leftarrow \hat{\epsilon}_\theta^{\text{cfg}} - \eta \cdot w(t+t_c)(\hat{\epsilon}_\theta(\hat{\mathbf{z}}_{t+t_c}, y, t+t_c) - \epsilon_\theta^{\textbf{SDU}})$  ▷ Score distillation using $\epsilon_\theta^{\textbf{SDU}}$

11:        $\hat{\mathbf{z}}_t \leftarrow \frac{1}{\sqrt{\alpha_{t+t_c}}}\left(\hat{\mathbf{z}}_{t+t_c} - \frac{\beta_{t+t_c}}{\sqrt{1-\bar{\alpha}_{t+t_c}}}\hat{\epsilon}_\theta^{\text{cfg}}\right) + \sqrt{\beta_{t+t_c}}\mathbf{n}$

12:        $\mathbf{n} \sim \mathcal{N}(\mathbf{0}, \mathbf{I})$ if $t > t_c$, else $\mathbf{n} = \mathbf{0}$

13:     **end if**

14: **end for**

15: **Output:** $\hat{\mathbf{z}}_0$

---

However, directly obtaining conditional samples from the original model, $p_\theta(\hat{\mathbf{z}}_t|y)$, is infeasible since the condition $y$ is encrypted. Instead, we leverage the unconditional distribution $p_\theta(\mathbf{z}_t)$, where $\mathbf{z}_t$ represents the latent vector from unconditional sampling, initialized as $\mathbf{z}_T = \hat{\mathbf{z}}_T$. This approach offers several advantages. First, sampling from $p_\theta(\mathbf{z}_t)$ is computationally more efficient than from $\hat{p}_\theta(\hat{\mathbf{z}}_t|y)$, as it avoids the complexities of handling the encrypted condition. Additionally, it can be computed using the original model with full steps, eliminating approximation errors. Thus, we target to minimize $D_{\text{KL}}(\hat{p}_\theta(\mathbf{z}_t|y), p_\theta(\mathbf{z}_t))$. Note that, since unconditional generation starts from the same noise as conditional generation, we assume $\mathbf{z}_t = \hat{\mathbf{z}}_t$ for the larger timesteps $t$.

According to our learning objective, the gradient of $\epsilon_\theta(\hat{\mathbf{z}}_t, y, t)$ is computed as:

$$\nabla_{\epsilon_\theta} D_{\text{KL}}(p_\theta(\mathbf{z}_t), \hat{p}_\theta(\hat{\mathbf{z}}_t|y)) = \mathbb{E}_{t,\epsilon}\left[w(t)\left(\hat{\epsilon}_\theta(\hat{\mathbf{z}}_t, y, t) - \epsilon_\theta(\mathbf{z}_t, t)\right)\right], \quad (7)$$

with the detailed derivation provided in Appendix B. The correction is applied as follows:

$$\hat{\epsilon}_\theta(\hat{\mathbf{z}}_t, y, t) \leftarrow \hat{\epsilon}_\theta(\hat{\mathbf{z}}_t, y, t) - \eta \cdot w(t)\left(\hat{\epsilon}_\theta(\hat{\mathbf{z}}_t, y, t) - \epsilon_\theta(\mathbf{z}_t, t)\right), \quad (8)$$

where $\eta$ is the learning rate. However, acquiring $\epsilon_\theta(\mathbf{z}_t, t)$ is challenging due to the differing number of denoising steps between unconditional and conditional generation. Specifically, let the timesteps for unconditional generation be $\{T, T-t_u, T-2\cdot t_u, \ldots, 0\}$, and for conditional generation, $\{T, T-t_c, T-2\cdot t_c, \ldots, 0\}$. Here, $t_c = m \cdot t_u$ for some $m \in \mathbb{N}$, as unconditional sampling uses more steps. At a given timestep $T - k \cdot t_c$ where $k \in \mathbb{N}$, our objective is to compute $\epsilon_\theta(\mathbf{z}_{T-k\cdot t_c})$ for the previous latent vector $\mathbf{z}_{T-(k+1)\cdot t_c}$, using all predictions between $T - k \cdot t_c$ and $T - (k+1) \cdot t_c$. Note that, for simplicity, we omit the timestep inputs for $\epsilon$ prediction.

To incorporate all predictions, we estimate $\epsilon_\theta(\mathbf{z}_t, t)$ from $\mathbf{z}_{T-k\cdot t_c}$ and $\mathbf{z}_{T-(k+1)\cdot t_c}$ by reverse-engineering the reverse process. Since $\mathbf{z}_{T-(k+1)\cdot t_c}$ is derived from all preceding predictions, its inclusion allows us to implicitly exploit information from these earlier steps. For DDPM (Ho et al., 2020), this estimation can be derived from Equation 2 as:

$$\epsilon_\theta^{\textbf{SDU}}\left(\mathbf{z}_{T-k\cdot t_c}, \mathbf{z}_{T-(k+1)\cdot t_c}\right) = \frac{\sqrt{1-\bar{\alpha}_t}}{\beta_t}\left(\mathbf{z}_{T-k\cdot t_c} - \sqrt{\alpha_t}\mathbf{z}_{T-(k+1)\cdot t_c}\right). \quad (9)$$

This approach can also be applied to other sampling methods, such as DDIM (Song et al.) through reverse engineering. The correction process is then updated as:

$$\hat{\epsilon}_\theta(\hat{\mathbf{z}}_t, y, t) \leftarrow \hat{\epsilon}_\theta(\hat{\mathbf{z}}_t, y, t) - \eta \cdot w(t)\left(\hat{\epsilon}_\theta(\hat{\mathbf{z}}_t, y, t) - \epsilon_\theta^{\textbf{SDU}}(\mathbf{z}_t, \mathbf{z}_{t-t_c})\right), \quad (10)$$

for a given timestep $t$. The full algorithm is outlined in Algorithm 1. Note that our score distillation method is compatible with both FHE and secure MPC. In practice, we use $\eta = 0.02$. To decrease the extent of distillation following the assumption, we adopt the weighting function as $w(t) = \frac{\alpha_t}{\sigma_t}$ where $\alpha_t = 1 - \beta_t$ and $\sigma_t = \sqrt{1-\alpha_t^2}$. During inference, we update $\hat{\epsilon}_\theta$ only once per timestep. The number of denoising steps in unconditional sampling is set to 50.

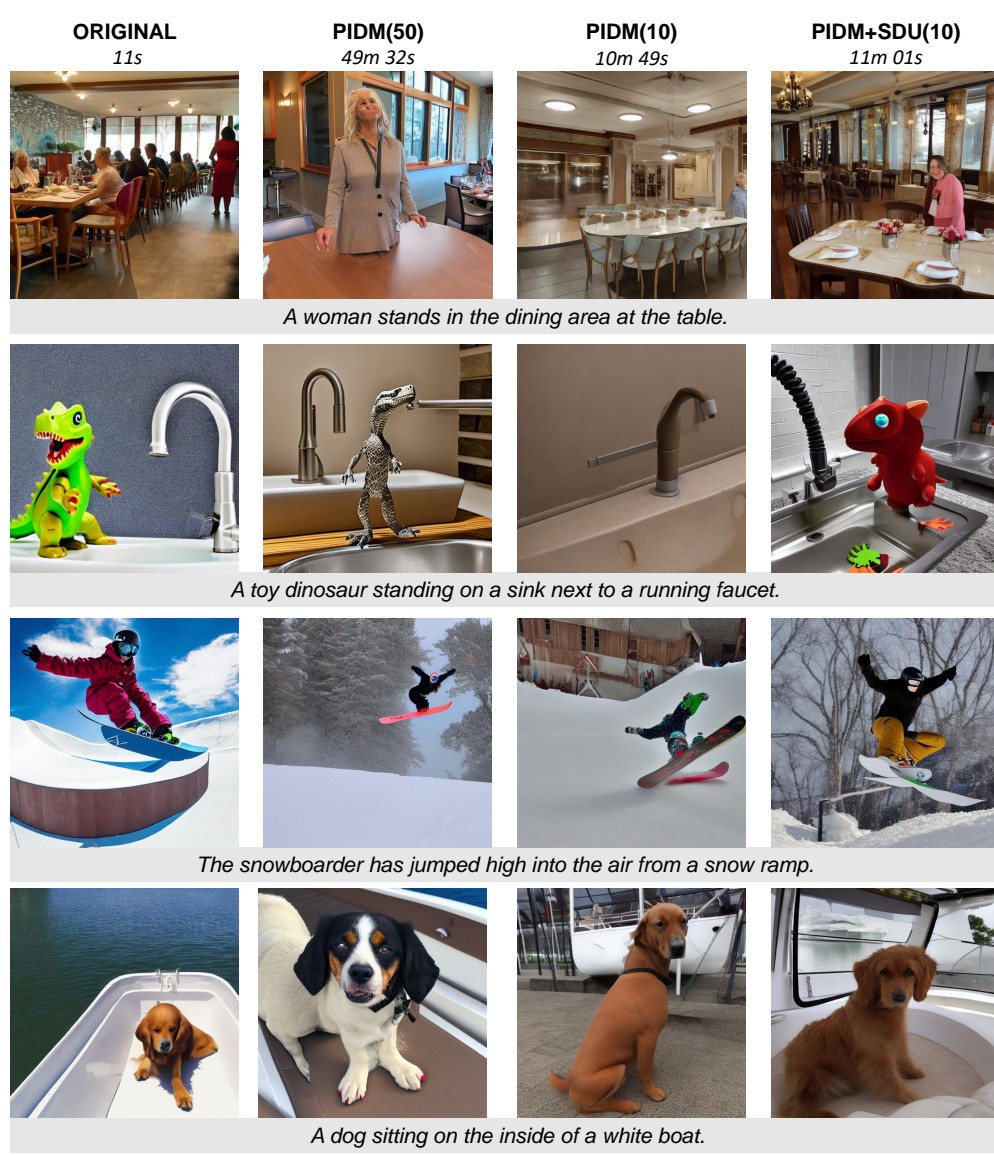

Figure 4: Comparison of text-to-image samples between Original SDv1.5, PIDM, PIDM+SDU.

# 4 EXPERIMENTS

In this section, we evaluate our approach on the text-to-image benchmark dataset in a secure multi-party computation (MPC) setting. We first briefly describe the experimental settings and evaluation protocol, followed by a qualitative comparison in Section 4.1 and a quantitative analysis in Section 4.2. Finally, we provide an analysis of the impact of varying the number of denoising steps in Section 4.3.

**Experimental setting** For the image generation model, we employ pre-trained Stable Diffusion v1.5 (Rombach et al., 2022) without any further fine-tuning. The DDIM sampler (Song et al.) is employed with a classifier-free guidance scale of 7.5 (Ho & Salimans, 2021). The evaluation of our method in the secure MPC setting is conducted using CrypTen (Knott et al., 2021). We follow the default environment of CrypTen with the exception of setting the precision to 22 bits and the number of iterations for the inverse square root to 10. To quantitatively assess the quality of text-to-image generation, we perform experiments on the MSCOCO (Lin et al., 2014).

Table 1: Comparison of FID, CLIP score, and latency on MSCOCO-30K.

| Method | FID ($\downarrow$) | CLIP-Score ($\uparrow$) | Latency ($\downarrow$) |
|---|---|---|---|
| SDv1.5 (50 step) | 13.25 | 0.3254 | 11s |
| PIDM (50 step) | 13.98 | **0.3215** | 49m 32s |
| PIDM (10 step) | 14.00 | 0.3174 | 10m 49s |
| PIDM+**SDU** (10 step) | **13.00** | 0.3181 | 11m 01s |

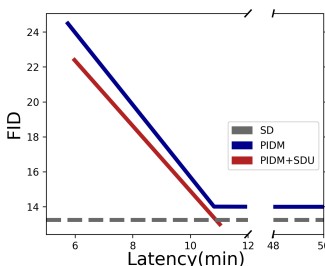

Figure 5: Denoising step analysis.

**Evaluation protocol**   We follow the evaluation protocol used in Kang et al. (2023). We use Frechet Inception Distance (FID) (Heusel et al., 2017) to measure image quality and the CLIP-Score (Radford et al., 2021) to evaluate alignments between text prompts and corresponding generated images. Specifically, we generate 30,000 images to compute both FID and CLIP-Score. To estimate the latency, we measure the wall clock time for MSCOCO by averaging the next ten repetitions after discarding the first image generation runs using a machine with Intel Xeon Platinum 8468 processors, 2TB RAM, and NVIDIA H100 (80GB VRAM). Due to the substantial computational overhead in secure MPC environments, combined with limited computational resources, we compute FID and CLIP-Score without using the secure MPC setting. Instead, we use a model where non-linear operations are approximated with the techniques employed in CrypTen and our approach, rather than performing evaluations in the secure MPC setting. For qualitative analysis and latency measurements, we evaluate our method in the secure MPC environment using CrypTen.

**Model notation**   SDv1.5 refers to the original diffusion model. PIDM represents the diffusion model where all non-linear operations are approximated in a secure MPC setting. PIDM+SDU denotes PIDM utilizing our score distillation from unconditional generation to sample images. ($k$ step) indicates that the model generates images using $k$ denoising steps.

### 4.1 QUALITATIVE ANALYSIS

To demonstrate the effectiveness of our method, we compare generated images from various models in Figure 6. We observe that PIDM (50 steps) produces high-quality images that are well-aligned with the text prompts, with image quality comparable to SDv1.5 (50 steps). This result implies that our approximation is effective. However, as the number of denoising steps is reduced, the image quality degrades, as seen when comparing PIDM (50 steps) and PIDM (10 steps). The images generated by PIDM (10 steps) exhibit blurred details. In contrast, when our method (SDU) is used during the sampling process, higher-quality images are produced, as demonstrated in the comparison between PIDM (10 steps) and PIDM+SDU (10 steps). The details and object boundaries in PIDM+SDU (10 steps) are noticeably clearer than those in PIDM (10 steps). Furthermore, even when comparing PIDM+SDU (10 steps) to PIDM (50 steps), the quality of PIDM+SDU (10 steps) is comparable to PIDM (50 steps) except for some image details.

### 4.2 QUANTITATIVE ANALYSIS

We also conduct a quantitative study in MSCOCO in Table 1 and the results are consistent with the qualitative result. We observe that PIDM (50 steps) is slightly inferior to SDv1.5 (50 steps) in both FID and CLIP-Score, indicating that our approximation techniques work. PIDM (10 steps) shows comparable performance to PIDM (50 steps). Surprisingly, PIDM+SDU (10 steps) improves the FID of PIDM (10 steps), achieving performance comparable to SDv1.5 (50 steps). This result implies that our score distillation approach effectively brings the score distribution closer to the true score distribution. Regarding alignment, since our score distillation method is not specifically designed to enhance alignment between text prompts and corresponding images, PIDM+SDU (10 steps) demonstrates a result comparable to that of PIDM (10 steps) in CLIP-Score.

For the latency, SDv1.5 (50 steps) is approximately 270 times faster than PIDM (50 steps), primarily due to the communication costs in the secure MPC setting and iterative approximations. However, latency can be significantly reduced by using fewer steps, as demonstrated by PIDM (10 steps). For

SDU, since unconditional generation occurs in the plaintext space, the additional latency introduced by our score distillation method is very small compared to the overall latency of PIDM (10 steps). The overall results indicate that our score distillation is both effective and efficient, yielding significant performance gains with only a small increase in latency.

### 4.3 ANALYSIS OF THE NUMBER OF DENOISING STEPS

Although reducing the number of sampling steps significantly decreases latency, it also leads to performance degradation. To address this, we investigate how many denoising steps can be reduced while maintaining only a small performance trade-off. The trade-off between performance and latency is illustrated in Figure 5. We observe a significant drop in performance when the number of denoising steps falls below 10. Nevertheless, our score distillation method still improves performance, even with fewer steps.

## 5 RELATED WORK

### 5.1 CONDITIONAL DIFFUSION MODEL

With the development of diffusion-based models(Sohl-Dickstein et al., 2015; Ho et al., 2020; Song et al.; Nichol & Dhariwal, 2021), significant progress has been made in various conditional image generation tasks, such as text-conditioned image synthesis (Saharia et al., 2022) and masked-region image editing (Lugmayr et al., 2022). Various open source models and commercial models such as DALLE (Ramesh et al., 2022), Imagen (Saharia et al., 2022), and GLIDE (Nichol et al., 2022) have contributed to this development. Recently, classifier-free guidance (Ho & Salimans, 2021) simplifies the conditional image generation process by directly incorporating class conditions into the diffusion model, eliminating the need for an external classifier. Based on this work, stable diffusion models (Rombach et al., 2022) have excelled in generating high-quality images efficiently, leading to the exploration of various applications.

In addition to these advancements, Score Distillation Sampling (SDS) (Poole et al.) which distills knowledge from a large pre-trained diffusion model into a more efficient student model further enhances the quality of generated images. Our approach shares similarities with SDS in that it transfers knowledge to conditional generation (student) from unconditional generation (teacher), but there are several key differences. First, we use score distillation during the inference phase, with a different input ordering of the KL divergence to reduce divergence in high-density samples. Additionally, our method takes more steps in the distillation process for unconditional generation than for conditional generation, and we further refine the score through reverse engineering.

### 5.2 PRIVATE INFERENCE

Fully homomorphic encryption (Gentry, 2009; Cheon et al., 2017) and secure multi-party computation (Yao, 1986; Evans et al., 2018; Damgård et al., 2019; Goldreich et al., 2019) are widely employed in private inference. CryptoNets (Gilad-Bachrach et al., 2016) pioneered the use homomorphic encryption for small neural networks in image classification using polynomial approximations. To reduce the latency, GAZELLE (Juvekar et al., 2018) proposes a hybrid method combining homomorphic encryption and two-party computation. Cheetah (Reagen et al., 2021) further accelerated private inference through hardware optimizations. For better approximations of non-linear operations like ReLU (Lou & Jiang, 2019; Ghodsi et al., 2020; Jha et al., 2021; Kundu et al., 2023; Peng et al., 2023; Li et al., 2024), inverse square root (Panda, 2022), and softmax (Lee et al., 2023), several works have proposed approximating algorithms. For transformer architectures, various methods (Hao et al., 2022; Li et al., 2023; Zeng et al., 2023; Zhang et al., 2023; Wu et al., 2024; Zimerman et al., 2024) are proposed to effectively approximate and compute non-linear operations such as GeLU (Hendrycks & Gimpel, 2016), LayerNorm (Ba, 2016), and attention mechanism (Vaswani, 2017). For diffusion models, HE-Diffusion (Chen & Yan, 2024) enables privacy-preserving image generation using homomorphic encryption. However, there remains a risk of input exposure to the model provider when sharing intermediate representations. In contrast, our method ensures the privacy of both inputs and outputs during image generation.

## 6 CONCLUSION

In this paper, we propose a new framework for private inference of pre-trained diffusion models. We introduce the practical approximation techniques, which are more robust in the larger input values. Additionally, we significantly reduce the latency with a small performance trade-off by using fewer steps and incorporating our new score distillation method, where the score from unconditional generation is distilled into conditional generation during the sampling process. In experiments, we demonstrate that the diffusion model with our approximations can achieve performance comparable to the original model. Furthermore, our score distillation method enhances performance, approaching the quality of full denoising steps even with fewer sampling steps.

**Limitations** In this study, we enable diffusion models to generate high-quality images in private inference while substantially reducing latency. However, the absolute latency of a single forward pass remains too high for practical deployment in real-world applications. By unlocking the secure diffusion model for the first time, we believe this research paves the way for practical private inference in diffusion models.

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

## A   APPROXIMATION DETAILS

### A.1   THE SCALING FACTOR OF GROUPNORM AND LAYERNORM

For numerical stability, input tensors are divided by a scaling factor $d$ before being processed by GroupNorm (Wu & He, 2018) and LayerNorm (Ba, 2016). The denominator $d$ is determined based on the channel size of a group. Specifically, in the text encoder, we set $d = 8$ for LayerNorm. In the latent diffusion model, we use $d = \max(\sqrt{(64 \cdot 64 \cdot 30)/d_x}, 1)$ for GroupNorm, where $d_x$ denotes the dimension of the channels of each group, and $d = 1$ for LayerNorm. In the decoder of the compression model, we apply $d = \sqrt{(512 \cdot 512 \cdot 256)/d_x}$ for GroupNorm.

### A.2   OTHER MINOR TECHNIQUES

We apply several techniques to adjust the output values of basic non-linear functions. For the sigmoid function, the output is clamped within the range $0 \leq \text{sigmoid}(x) \leq 1$, and a scaling factor $d = 16$ is used when computing the reciprocal: $\text{reciprocal}(x/d)/d$. For the softmax function, we ensure that the sum of the output values over a given axis equals 1 after softmax processing. In GeLU, we apply a scaling factor $d = 64$ when computing the sigmoid function within the text encoder. Additionally, we increase the number of iterations to 20 for the exponential approximation in the sigmoid function used within GeLU and LayerNorm of the text encoder, as well as SiLU in the decoder.

## B   DERIVING SCORE DISTILLATION FROM UNCONDITIONAL GENERATION

We now derive the gradient of our objective function in Equation 8. We consider the KL divergence in Equation 6.

$$D_{\text{KL}}(p_\theta(\mathbf{z}_t), \hat{p}_\theta(\hat{\mathbf{z}}_t|y)) = \mathbb{E}_{t,\epsilon}\left[\log p_\theta(\mathbf{z}_t) - \log \hat{p}_\theta(\hat{\mathbf{z}}_t|y)\right]$$

$$\nabla_{\hat{\epsilon}_\theta} D_{\text{KL}}(p_\theta(\mathbf{z}_t), \hat{p}_\theta(\hat{\mathbf{z}}_t|y)) = \mathbb{E}_{t,\epsilon}\left[\nabla_{\hat{\epsilon}_\theta} \log p_\theta(\mathbf{z}_t) - \nabla_{\hat{\epsilon}_\theta} \log \hat{p}_\theta(\hat{\mathbf{z}}_t|y)\right]$$

The first term is computed using $\nabla_{\mathbf{z}_t} \log \hat{p}_\theta(\mathbf{z}_t) \approx s_\theta(\mathbf{z}_t)$ and the assumption that $\mathbf{z}_t = \hat{\mathbf{z}}_t$:

$$\nabla_{\hat{\epsilon}_\theta} \log p_\theta(\mathbf{z}_t) = s_\theta(\mathbf{z}_t)\frac{\partial \mathbf{z}_t}{\partial \epsilon_\theta} = -\frac{1}{\sqrt{\beta_t}}\epsilon_\theta(\mathbf{z}_t)\frac{\partial \mathbf{z}_t}{\partial \hat{\epsilon}_\theta} = -\frac{1}{\sqrt{\beta_t}}\epsilon_\theta(\mathbf{z}_t)\frac{\partial \hat{\mathbf{z}}_t}{\partial \hat{\epsilon}_\theta} = -\epsilon_\theta(\mathbf{z}_t).$$

Similarly, we compute $\nabla_{\hat{\epsilon}_\theta} \log \hat{p}_\theta(\hat{\mathbf{z}}_t|y)$ as:

$$\nabla_{\hat{\epsilon}_\theta} \log \hat{p}_\theta(\hat{\mathbf{z}}_t|y) = s_\theta(\hat{\mathbf{z}}_t|y)\frac{\partial \hat{\mathbf{z}}_t}{\partial \hat{\epsilon}_\theta} = -\hat{\epsilon}_\theta(\hat{\mathbf{z}}_t|y).$$

Combining the previous equations, the gradient of $\nabla_{\hat{\epsilon}_\theta} D_{\text{KL}}(p_\theta(\mathbf{z}_t), \hat{p}_\theta(\hat{\mathbf{z}}_t|y))$ is calculated by:

$$\nabla_{\hat{\epsilon}_\theta} D_{\text{KL}}(p_\theta(\mathbf{z}_t), \hat{p}_\theta(\hat{\mathbf{z}}_t|y)) = \mathbb{E}_{t,\epsilon}\left[w(t)\left(\hat{\epsilon}_\theta(\hat{\mathbf{z}}_t, y, t) - \epsilon_\theta(\mathbf{z}_t, t)\right)\right].$$

## C   MORE EXAMPLES

Below, we provide visualizations of generation examples for extended qualitative assessment. As shown in the following examples, the use of SDU at the same denoising step, compared to PIDM, results in superior image quality and more accurate reflection of the textual content. Furthermore, when compared to PIDM with 50 steps, it is observed that comparable image quality can be achieved with only one-fifth of the number of steps.

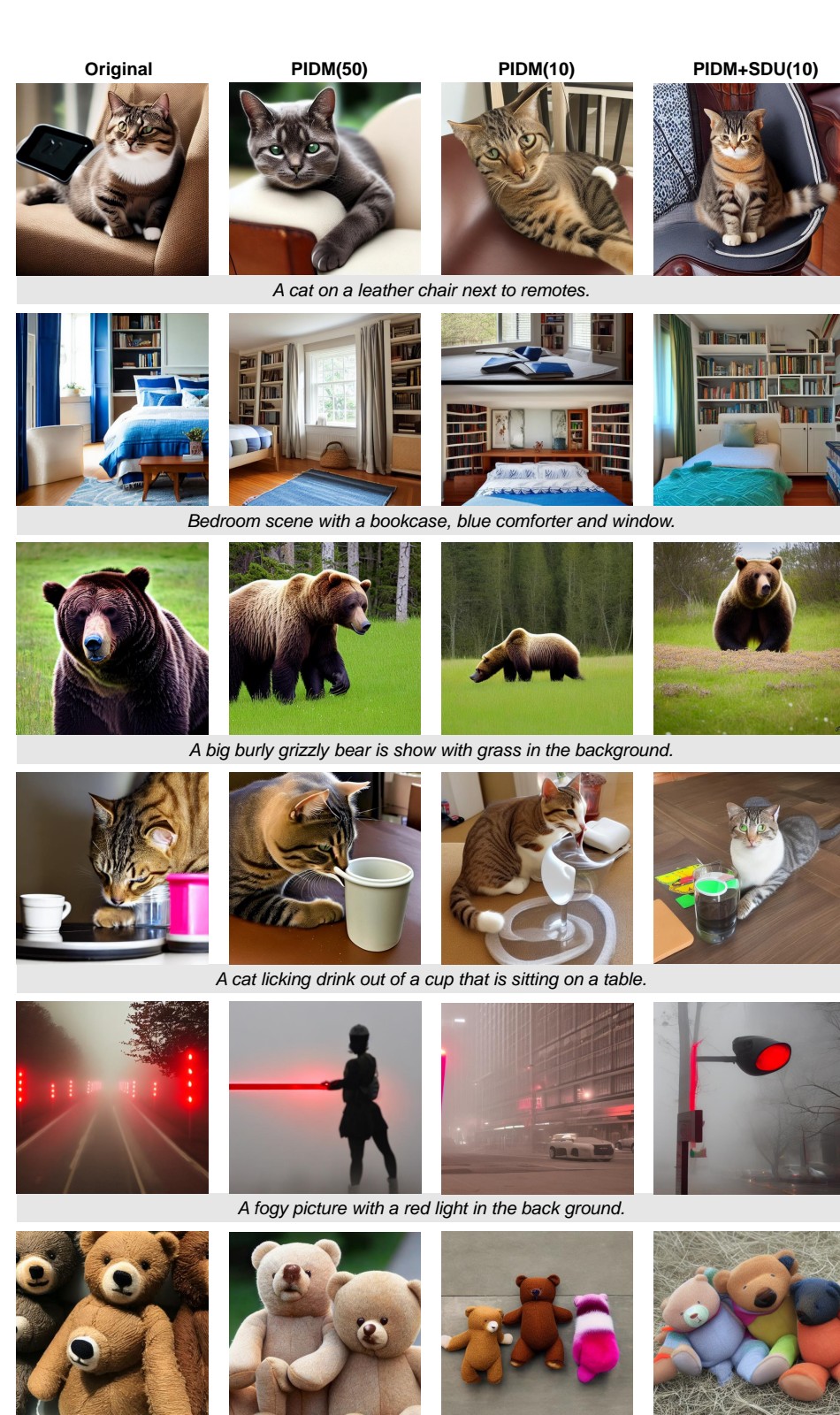

Figure 6: Comparison of text-to-image samples between Original SDv1.5, PIDM, PIDM+SDU.

