# OpenReview forum: "Secure Diffusion Model Unlocked: Efficient Inference via Score Distillation"
_ICLR.cc/2025/Conference — ICLR 2025 Conference Withdrawn Submission_

### Official Review · Reviewer_Jfsm · 2024-10-29

**Soundness:** 3
**Presentation:** 3
**Contribution:** 2
**Rating:** 3
**Confidence:** 3

**Summary:**

This paper proposes a new framework for privacy-preserving diffusion model inference. The authors introduce a new method to handle non-linear computations in diffusion models and reduce the number of denoising steps for faster inference. Finally, they conduct empirical studies to validate the effectiveness of the proposed method.

**Strengths:**

The main advantages can be listed as follows:

1.	The paper adopts several techniques for non-linear operations and proposes a new score distillation sampling method to reduce the computational overhead during private diffusion model inference.
2.	The paper conducts empirical studies on Stable Diffusion v1.5 to show the effectiveness of the proposed method.

**Weaknesses:**

Despite the strengths, there are some issues with the paper as follows:

1.	I am concerned about the novelty of this paper. The techniques for polynomial approximation of nonlinear functions in Section 3.1 appear to have been proposed in previous studies. For instance, [1] also employs the hyperbolic tangent function to approximate GeLU. The only new contribution in Section 3.1 seems to be the scaling factor, $\sqrt{d}$.
2.	I am also concerned about the validity of the experimental evaluation of the proposed method. The authors conducted quantitative analysis on only one model and one benchmark dataset without comparison to other methods, as shown in Table 1. Including comparisons across more datasets and models would significantly enhance the credibility and robustness of the experimental results.

[1] Iron: Private inference on transformers. NeurIPS, 2022.

**Questions:**

1.	There seems to be a hyperparameter $m$ in Algorithm 1. What are the hyperparameters of the proposed method, and how are they configured to achieve the trade-off between latency and performance?
2.	How robust is the proposed framework across different types of diffusion models?

---

> ### Author Response · Authors · 2024-11-20
> **Response to Reviewer Jfsm**
>
> We appreciate your insightful comments.
>
> ---
>
> ### [W1] The techniques for polynomial approximation of nonlinear functions in Section 3.1 appear to have been proposed in previous studies.
> We agree that each approximation trick is not novel in a general context. However, our intention is not to propose superior approximation methods for each operation in a general context. Instead, we aim to introduce straightforward approximation techniques specifically tailored for private inference within conditional diffusion models (~1.45B parameters), which entail extensive computations across a wide range of values. Given this challenge, we focus on proposing stable approximations that operate effectively over short value ranges with feasible outcomes.
>
> ---
>
> ### [W2] I am also concerned about the validity of the experimental evaluation of the proposed method.
> To further strengthen our paper, we will add the validation under other architectures with other datasets in the revision. Nevertheless, it is important to emphasize that our current experiments were conducted using MSCOCO-30K, a widely used benchmark dataset for zero-shot text-to-image generation, and Stable Diffusion, a large, standard model that is widely adopted in this field. This ensures that **our initial validation was performed under a well-established and representative setting [1-5]**.
>
> [1] Ramesh, Aditya, et al. "Zero-shot text-to-image generation." *International conference on machine learning*. *ICML*, 2021.
>
> [2] Rombach, Robin, et al. "High-resolution image synthesis with latent diffusion models." *CVPR,* 2022.
>
> [3] Saharia, Chitwan, et al. "Photorealistic text-to-image diffusion models with deep language understanding." *NeurIPS*, 2022.
>
> [4] Kang, Minguk, et al. "Scaling up gans for text-to-image synthesis." *CVPR,* 2023.
>
> [5] Chen, Junsong, et al. "PixArt-$\alpha$: Fast Training of Diffusion Transformer for Photorealistic Text-to-Image Synthesis." *ICLR,* 2024.
>
> ---
>
> ### [Q1] What are the hyperparameters of the proposed method, and how are they configured to achieve the trade-off between latency and performance?
> Hyperparameters of SDU are suggested in lines 320-323. **The trade-off between latency and performance is only associated with the number of steps in conditional generation**, while other hyperparameters have negligible impact on latency and affect only performance. The trade-off with respect to the number of conditional generation steps is illustrated in Figure 5, where performance is preserved up to 10 steps. However, reducing the number of steps further results in performance degradation.
>
> Specifically, for $m$, our goal is to **correct the scores of conditional generation to align with the distribution of the original model**. To achieve this, **we set $m$ to satisfy that the number of steps used in unconditional generation is equal to the number of steps used in the original model**. For example, if the number of steps used for conditional generation is 10 and the original model uses 50 steps, we set $m=5$. Similarly, if the number of steps for conditional generation is 5, we set $m=10$. For the trade-off with respect to $m$, there is almost no trade-off between performance and latency according to $m$ since $m$ does not affect latency significantly. Specifically, the operations for unconditional generation are performed in plaintext space, making them nearly 60 times faster than conditional generation as shown in Table 1. Moreover, since these operations are independent of the user input, they can be precomputed offline. As a result, **the impact of $m$ on latency is negligible**. However, if $m$ is set to induce that the number of steps used in unconditional generation is smaller than the number of steps used in the original model, distortion in the score values can occur, potentially leading to slight performance degradation. Therefore, since the latency impact of $m$ is negligible compared to the latency of conditional generation, we recommend setting $m$ to achieve that **the number of steps used in unconditional generation is equal to the number of steps used in the original model** to maintain consistency and avoid performance loss.
>
> ---
>
> ### [Q2] How robust is the proposed framework across different types of diffusion models?
> While the scale of approximations may vary depending on the model, SDU adopts a noise correction approach during the sampling process, making it compatible with any diffusion model architecture. To further enhance our paper, we will add the validation under other architectures in the revision.

---

### Official Review · Reviewer_2abM · 2024-11-04

**Soundness:** 2
**Presentation:** 2
**Contribution:** 2
**Rating:** 5
**Confidence:** 2

**Summary:**

The paper proposes running secure inference from a latent diffusion model, in a way that reduces the amount of computation that has to be done in ciphertext space (that is either in MPC or under homomorphic encryption).

They describe this approach, describe some non-linear approximations they had to make and display some results of experiments done on this.

**Strengths:**

The problem makes sense as something to want to improve, shifting computation out of the encrypted space makes sense as an approach to do that.

The paper proposes some new seemingly improved approximations for non-linear functions in MPC.
The new non-linear approximations may be an improvement though I can't really understand whether they are appropriate and good without knowing the context better (see weaknesses) and it isn't justified that it constitutes a substantial contribution.

**Weaknesses:**

I can't follow what this paper is proposing, this may well be due to my lack of familiarity with diffusion models but I think the presentation is also confusing and that isn't helping.

The preliminaries define epsilon_theta in terms of epsilon_theta itself and epsilon which is never defined (though I infer from the expectation in (3) being taken over it that it is a random variable). I assume 'the epsilon prediction' is to be minimised though it isn't clear what it is as it depends on epsilon. epsilon_theta seems to be a vector when being taken off x would I therefor be right in thinking that the score function is vector valued?

Lines 9 and 10 in Algorithm 1 depend on an \hat{\epsilon_\theta} that isn't defined anywhere except possibly in (7) and (8) in terms of itself, though these seem to be describing corrections to some baseline that isn't defined.

There is no novelty specifically in secure computation itself here, though if the work can be substantially moved to the clear that would be valuable.

**Questions:**

I don't think I really have questions, just the above points on presentation that I think should be clarified and openness to the possibility that a reviewer with more knowledge of notation in diffusion model papers can identify understandable substantive novelty here.

---

> ### Author Response · Authors · 2024-11-20
> **Response to Reviewer 2abM**
>
> We appreciate your constructive feedback.
>
> ---
>
> ### [W1] The presentation is also confusing and that isn't helping.
> We apologize for the unclear definitions. $\epsilon_\theta$ refers to the noise vector predicted by the model based on the input. Specifically, when diffusion models were first proposed, they predicted the mean ($\mu_\theta$) of the latent variable at timestep $t-1$ using the latent variable at the next timestep $t$. However, this approach yielded inferior performance compared to GANs. In DDPM, it was shown that by utilizing the forward process, $\mu_\theta$ can be expressed in terms of $x_0$ and noise ($\epsilon$). Furthermore, minimizing the loss to match $\Vert \mu_\theta - \mu\Vert$ is equivalent—except for a constant scale—to predicting the noise $\Vert\epsilon_\theta - \epsilon\Vert$ directly. Training the model this way enables diffusion models to achieve performance comparable to GANs. In this context, $\epsilon_\theta$ is introduced as the noise prediction of the model. For $\hat \epsilon_\theta $, this indicates the noise predicted by the approximated models. To discriminate it from the prediction of the original model ($\epsilon_\theta$), we use $\hat{\epsilon}_\theta$. We will add definitions of them in the revision.
>
> ---
>
> ### [W2] There is no novelty specifically in secure computation itself here.
> We agree that each approximation trick is not novel in a general context. However, our intention is not to propose superior approximation methods for each operation in a general context. Instead, we aim to introduce straightforward approximation techniques specifically tailored for private inference within conditional diffusion models (~1.45B parameters), which entail extensive computations across a wide range of values. Given this challenge, we focus on proposing stable approximations that operate effectively over short value ranges with feasible outcomes.

---

### Official Review · Reviewer_uDxY · 2024-11-08

**Soundness:** 1
**Presentation:** 1
**Contribution:** 1
**Rating:** 1
**Confidence:** 4

**Summary:**

Secure Diffusion introduces an approximation function and employs a Score Distillation method, which utilizes the score of an unconditional generation path in an unencrypted space to efficiently handle encrypted inputs. This approach reduces computational load in encrypted data environments while maintaining quality similar to the original model, achieving faster inference speeds.

**Strengths:**

- A new method called SDU is introduced.

- For model comparison, it is necessary to derive the distribution $p(z|y)$ using the encrypted condition $y$. The idea here is that this can be approximated by using randomly sampled $p(z)$, which provides a similar outcome.

**Weaknesses:**

- The approximation function was only compared with CrypTen, an open-source library, without any comparisons to approximation functions used in recent papers. (CrypTen was published in 2021, but it’s unclear how up-to-date its approximation methods are in this paper.) These are some examples of the recent papers.

* Qu, Hongyuan, and Guangwu Xu. "Improvements of Homomorphic Secure Evaluation of Inverse Square Root." International Conference on Information and Communications Security. Singapore: Springer Nature Singapore, 2023.
* Luo, Jinglong, et al. "SecFormer: Fast and Accurate Privacy-Preserving Inference for Transformer Models via SMPC." Findings of the Association for Computational Linguistics ACL 2024. 2024.
* Lu, Wen-jie, et al. "Bumblebee: Secure two-party inference framework for large transformers." Cryptology ePrint Archive (2023).

- They point out that the main limitation of existing approximation functions is that they fail outside certain ranges. Their contribution is adding simple scaling and other nonlinear functions, like tanh, to improve stability for larger inputs.

- However, there is no analysis of the approximation accuracy of their proposed function.

- The experiments are very limited. There is no comparison with prior research, and the results consist only of a single chart comparing their method with the score of the original model.

**Overall review**

There were many unclear points in the scenario, making it difficult to understand.

Ultimately, the model proceeds as follows:
1. Encrypt the original diffusion model using an approximation function.
2. Apply Score Distillation to align the encrypted model’s output with the original model’s output based on encrypted inputs.
3. Return the result.

The idea in step 2—approximating via Score Distillation without obtaining the original model’s result on encrypted inputs—seems promising. However, the paper does not show any experiments to illustrate the efficiency of this idea.

The two main contributions are improvements in approximation functions and Score Distillation. However, the approximation function appears poorly developed, as it seems they did not follow up on the latest research in approximation functions.

**Questions:**

1. it is necessary to measure performance in an MPC environment rather than in a plaintext setting.
2. In the presented experiment, FID, CLIP-Score, and latency were measured. However, without a baseline for comparison, it may be difficult for users to intuitively understand the efficiency of the proposed PIDM based on the numbers alone. Demonstrating different FID, CLIP-Score, and latency results using other standard models, such as HE-Diffusion (Chen & Yan, 2024) or a model other than SDv1.5, under the same conditions may help substantiate the usefulness of PIDM.
3. It was mentioned that GeLU includes both the sigmoid and Gaussian error functions, but the GELU function does not actually include the sigmoid. How can you explain this?
4. A cubic approximation using tanh was presented, which is a method proposed in Gaussian Error Linear Units (GELUs) by Dan Hendrycks and Kevin Gimpel, so a citation for this would be appropriate.
* Hendrycks, Dan, and Kevin Gimpel. "Gaussian error linear units (gelus)." arXiv preprint arXiv:1606.08415 (2016).
5. For Reciprocal, LayerNorm, GroupNorm, and Softmax, it was mentioned that previous approximation functions had range issues, and scaling was applied to increase this range. Increasing the approximation range via scaling is a very common method. However, if the range is scaled by a factor of d, the impact of error in the approximation function is proportionally increased by d. Therefore, approximation function research typically aims to achieve the highest accuracy within a limited degree, rather than merely extending the range. An analysis on this aspect seems necessary.
6. An accurate analysis of the accuracy of the approximation functions in relation to the number of computations and the feasible approximation range seems necessary. The need for a wide approximation range has not been mentioned.
7. For the approximation of erf, it was only mentioned that crypten used a Taylor approximation, but the degree used was not specified. Furthermore, there was no mention of which approximation methods were applied to other non-linear functions by crypten. An analysis of the approximation methods and computation count used in the library should be included.
8. In Appendix B, it is mentioned that 20 iterations are used for exp in the GeLU approximation. However, 20 iterations cannot achieve a stable approximation within the range of [-10, 10], as shown in figure 2b. Additional clarification on the number of iterations used in figure 2b would be helpful.
9. It is unclear why the encrypted model was not tuned to match the original model from the beginning. The approach involves encrypting a model trained on public data, then adjusting for error accuracy with the user’s encrypted inputs. This raises a fundamental question: if the encrypted model had been pre-tuned to match the original model’s error accuracy on public data, wouldn’t techniques like Score Distillation on encrypted inputs be unnecessary? Please clarify why they chose this approach over pre-tuning the encrypted model.

---

> ### Author Response · Authors · 2024-11-20
> **Response to Reviewer uDxY (1/2)**
>
> We appreciate your valuable comments.
>
> ---
>
> ### [W1-3, Q3-8] Concerns about approximation tricks.
> Thank you for your detailed and insightful feedback. We agree that an in-depth analysis incorporating recent approximation approaches would make our work solid and strengthen our contributions. However, our intention is not to propose superior approximation methods for each operation in a general context. Instead, we aim to introduce straightforward approximation techniques specifically tailored for private inference within conditional diffusion models (~1.45B parameters), which entail extensive computations across a wide range of values. Given this challenge, we focus on proposing stable approximations that operate effectively over short value ranges with feasible outcomes.
>
> ---
>
> ### [W4, Q2] The experiments are very limited. There is no comparison with prior research.
> **To the best of our knowledge, our work is the first** **to propose private inference on Text-to-Image diffusion models while preserving input privacy through encryption, so there is no prior work for direct comparison**. For **HE-Diffusion**, as mentioned in the main paper, the model provides intermediate vectors from the middle layers to the server without encryption. This setup removes the need for approximation, so the performance drop would be small. However, it has a critical limitation—**a privacy vulnerability from inversion attacks**. For this reason, we did not include HE-Diffusion as a baseline, since comparison would not be fair. Instead, to demonstrate the effectiveness of our main contribution, SDU, we included varying own baselines. As upper bounds, we used the original model (50 steps) and an approximated model (50 steps). We also presented the performance of the approximated model (10 steps) without SDU as a comparative baseline.
>
> For evaluation datasets, to further strengthen our paper, we will add the validation under other architectures with other datasets in the revision. Nevertheless, it is important to emphasize that our current experiments were conducted using MSCOCO-30K, the most widely used benchmark dataset for zero-shot text-to-image generation, and Stable Diffusion, a large, standard model that is widely adopted in this field. This ensures that **our initial validation was performed under a well-established and representative setting [1-5]**.
>
> [1] Ramesh, Aditya, et al. "Zero-shot text-to-image generation." *International conference on machine learning*. *ICML*, 2021.
>
> [2] Rombach, Robin, et al. "High-resolution image synthesis with latent diffusion models." *CVPR,* 2022.
>
> [3] Saharia, Chitwan, et al. "Photorealistic text-to-image diffusion models with deep language understanding." *NeurIPS*, 2022.
>
> [4] Kang, Minguk, et al. "Scaling up gans for text-to-image synthesis." *CVPR,* 2023.
>
> [5] Chen, Junsong, et al. "PixArt-$\alpha$: Fast Training of Diffusion Transformer for Photorealistic Text-to-Image Synthesis." *ICLR,* 2024.
>
> ---
>
> ### [W5] The paper does not show any experiments to illustrate the efficiency of SDU.
> The efficiency of SDU is demonstrated in Table 1. Compared to the approximated model (50 steps), **our approach achieves nearly 5x faster inference, yet it maintains a superior FID score even compared to the original model**. This shows that SDU enhances the efficiency of the Stable Diffusion model by enabling effective performance with fewer denoising steps. Additionally, while our method requires an additional 10 seconds during inference due to unconditional generation compared to the approximated model (10 steps), this step is independent of the input conditions and can be precomputed offline. As a result, in actual online inference, this approach effectively reduces the number of denoising steps without incurring any significant additional cost.
>
> ---
>
> ### [Q1] It is necessary to measure performance in an MPC environment rather than in a plaintext setting.
> Since conducting evaluations on MSCOCO-30K in an MPC setting requires an extensive amount of computational resources, we performed quantitative evaluations on the approximated model without encryption and communication. To complement the gap, we conduct experiments in MPC settings for qualitative studies and latency measurements. The qualitative study results, shown in Figures 4 and 6, demonstrate that applying SDU produces higher-quality images compared to the approximated model (50 steps) in MPC settings.

---

> ### Author Response · Authors · 2024-11-20
> **Response to Reviewer uDxY (2/2)**
>
> ### [Q9] It is unclear why the encrypted model was not tuned to match the original model from the beginning.
>
> **Pre-tuning the encrypted model requires significant additional computational costs.** Specifically, SDv1.5 is a large model with **1.45B parameters** trained on the extensive LAION-400M dataset consisting of **400M text-image pairs [6]**. Pre-tuning such a model on this dataset requires considerable computational resources. In contrast, our SDU does not require additional fine-tuning and requires negligible additional computations since precomputations can be performed offline as mentioned in [W5]. Moreover, our method enables models to reduce denoising steps while improving performance, offering a significant advantage.
>
> [6] Schuhmann, Christoph, et al. "LAION-400M: Open Dataset of CLIP-Filtered 400 Million Image-Text Pairs." *NeurIPS Workshop*, 2021.

---

### Note · Authors · 2024-11-20

I have read and agree with the venue's withdrawal policy on behalf of myself and my co-authors.